# EFFICIENT EXPLORATION USING MODEL-BASED QUALITY-DIVERSITY WITH GRADIENTS

## ABSTRACT

Exploration is a key challenge in Reinforcement Learning, especially in long-horizon, deceptive and sparse-reward environments. For such applications, population-based approaches have proven effective. Methods such as Quality-Diversity deals with this by encouraging novel solutions and producing a diversity of behaviours. However, these methods are driven by either undirected sampling (i.e. mutations) or use approximated gradients (i.e. Evolution Strategies) in the parameter space, which makes them highly sample-inefficient. In this paper, we propose a model-based Quality-Diversity approach. It extends existing QD methods to use gradients for efficient exploitation and leverage perturbations in imagination for efficient exploration. Our approach optimizes all members of a population simultaneously to maintain both performance and diversity efficiently by leveraging the effectiveness of QD algorithms as good data generators to train deep models. We demonstrate that it maintains the divergent search capabilities of population-based approaches on tasks with deceptive rewards while significantly improving their sample efficiency and quality of solutions.

## 1 INTRODUCTION

Reinforcement Learning (RL) has demonstrated tremendous abilities to learn challenging tasks across a range of applications (Mnih et al., 2015; Silver et al., 2016; Akkaya et al., 2019). However, they generally struggle with exploration as the agent can only gather data by interacting with the environment. On the other hand, population based learning methods have shown to be very effective approaches (Jaderberg et al., 2017; Vinyals et al., 2019; Ecoffet et al., 2021; Wang et al., 2020). In contrast to single agent learning, training a population of agents allow diverse behaviors and data to be collected. This results in exploration that can better handle sparse and deceptive rewards Ecoffet et al. (2021) as well as alleviate catastrophic forgetting (Conti et al., 2018).

An effective way to use the population of agents for exploration are novelty search methods (Lehman & Stanley, 2011a; Conti et al., 2018) where the novelty of the behaviors of new agents is measured with respect to the population. This novelty measure is then used in place of the conventional task reward similar to curiosity and intrinsic motivation approaches (Oudeyer et al., 2007; Bellemare et al., 2016; Pathak et al., 2017). Quality-Diversity (QD) Pugh et al. (2016); Cully et al. (2015); Chatzilygeroudis et al. (2021) extends this but also optimizes all members of the population on the task reward while maintaining the diversity through novelty. Beyond exploration, the creativity involved in finding various ways to solve a problem/task (i.e. the QD problem) is an interesting aspect of general intelligence that is also associated with adaptability. For instance, discovering diverse walking gaits can enable rapid adaptation to damage (Cully et al., 2015).

However, a drawback of conventional population based approaches is the large amounts of samples and evaluations required, usually in the order of millions. Some methods that utilize Evolutionary Strategies (ES) and more recently MAP-Elites Mouret & Clune (2015) (a common QD algorithm), sidestep this issue as they can parallelize and scale better with compute (Salimans et al., 2017; Conti et al., 2018; Lim et al., 2022) than their Deep RL counterparts, resulting in faster wall-clock times. Despite this, they still come at a cost of many samples. One of the main reasons for this lies in the underlying optimization operators. QD methods generally rely on undirected search methods such as objective-agnostic random perturbations (Mouret & Clune, 2015; Vassiliades & Mouret, 2018) to favor creativity and exploration. More directed search such as ES has also been used (Colas et al., 2020) but relies on a large number of such perturbations (∼thousands) to approximate a single step of natural gradient to direct the improvement of solutions.

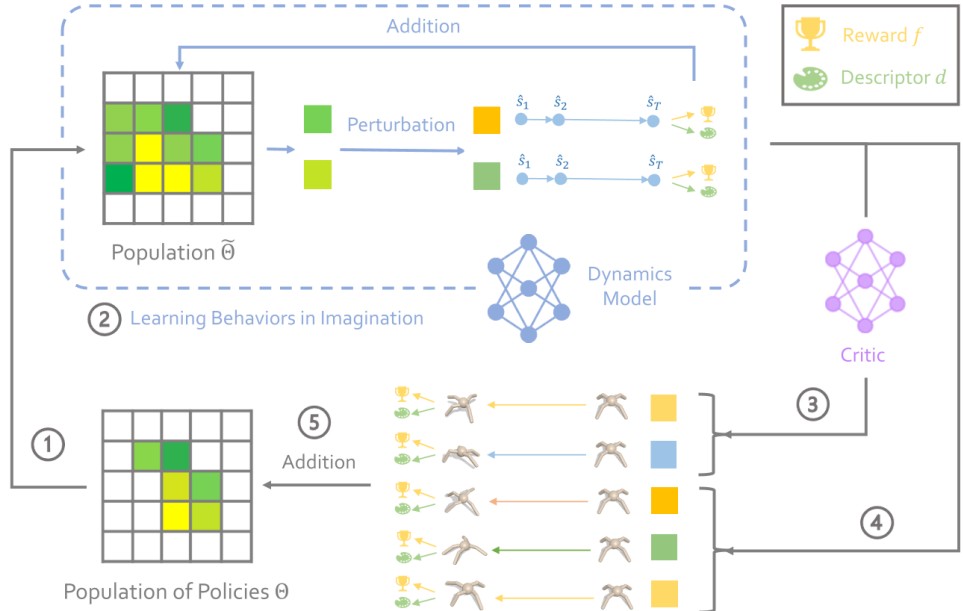

Figure 1: The GDA-QD algorithm can be summarized as follows: (1) the current population $\Theta$ is copied in $\tilde{\Theta}$, (2) $\tilde{\Theta}$ is used to perform multiple steps of QD optimization fully in imagination using the dynamics model, (3) the critic update sampled policies learned in imagination with policy-gradient updates; they are concatenated with (4) policies sampled from the resulting population learned in imagination $\tilde{\Theta}$, (5) these concatenated batch of policies are evaluated in the environment and used to update the real population of policies for the next optimization loop; the transitions collected in the environment are then used to train the dynamics model and the critic.

In this paper, we introduce an extended version of Dynamics-Aware QD (DA-QD-ext) as well as Gradient and Dynamics Aware QD (GDA-QD), a new model-based QD method to perform sample-efficient exploration in RL. GDA-QD optimizes an entire population of diverse policies through a QD process in imagination using a learned dynamics model. Additionally, GDA-QD augments the conventional QD optimization operators with policy gradient updates using a critic network to obtain a more performant population. Beyond the effective exploration capabilities of QD methods, they are also excellent data generators. We leverage this idea to harvest a diversity of transitions to train the dynamics model and the critic. Thus, GDA-QD combine the powerful function-approximation capabilities of deep neural networks with the directed-search abilities of gradient-based learning and the creativity of population-based approaches. We demonstrate that it successfully outperforms both Deep RL and QD baselines in a hard-exploration task. GDA-QD exceeds the performance of baseline QD algorithms by $\sim 1.5$ times, and can reach the same results in 5 times less samples.

## 2 PRELIMINARIES

### 2.1 REINFORCEMENT LEARNING

Reinforcement Learning (RL) is commonly formalised as a Markov Decision Process (MDP) (Sutton & Barto, 2018) represented by the tuple $(\mathcal{S}, \mathcal{A}, \mathcal{P}, \mathcal{R})$, where $\mathcal{S}$ and $\mathcal{A}$ are the set of states and actions. $\mathcal{P}(s_{t+1}|s_t, a_t)$ is the probability of transition from state $s_t$ to $s_{t+1}$ given an action $a_t$, where $s_t$, $s_{t+1} \in \mathcal{S}$ and $a_t \in \mathcal{A}$. The reward function defines the reward obtained at each timestep $r_t = r(s_t, a_t, s_{t+1})$ when transitioning from state $s_t$ to state $s_{t+1}$ under action $a_t$. An agent acting in the environment selects its next action based on the current state $s_t$ by following a policy $\pi_\theta(a_t|s_t)$. The conventional objective in RL is then to optimize the parameters $\theta$ of policy $\pi_\theta$, such that it maximizes the expected cumulative reward $R(\tau) = \sum_{t=1}^{T} r_t$ over the entire episode trajectory $\tau$:

$$J(\pi_\theta) = \mathbb{E}_{\tau \sim \pi_\theta} [\mathcal{R}(\tau)] \tag{1}$$

---

**Algorithm 1:** DA-QD-ext and GDA-QD (highlighted lines are specific to GDA-QD)

---

1 **Inputs:** $J$ num iterations, $N$ num imagined iterations, $p_{gradient}$ prop. of gradient-updated policies, and $p_{model} = 1 - p_{gradient}$ (for DA-QD-ext $p_{gradient} = 0$ and $p_{model} = 1$)

2 **Initialisation:** $\Theta_0 \leftarrow$ init_population(), $q_\phi \leftarrow$ init_dynamics_model(), $Q_\psi \leftarrow$ init_critic()

3 **for** $j = 1, ..., J$ **do**

4     $\tilde{\Theta}_j \leftarrow \Theta_j$ // copy $\Theta_j$ in $\tilde{\Theta}_j$

5     // Optimize population in imagination

6     **for** $it_{imagination} = 1, ..., N$ **do**

7        $\theta \leftarrow$ random_selection($\tilde{\Theta}_j$)

8        $\tilde{\theta} \leftarrow$ perturb($\theta$)

9        $F(\tilde{\theta}), d(\tilde{\theta}) \leftarrow$ evaluate_imagination($\tilde{\theta}, q_\phi$)

10        $\tilde{\Theta}_j \leftarrow$ update_population($\tilde{\theta}, F(\tilde{\theta}), d(\tilde{\theta})$)

11     $\theta_{new} \leftarrow$ get_last_added($\tilde{\Theta}_j$) // get last policies added to $\tilde{\Theta}_j$

12     $\theta_{model} \leftarrow$ select($\theta_{new}, p_{model}$)

13     $\theta_{gradient} \leftarrow$ apply_gradient($Q_\psi$, select($\theta_{new}, p_{gradient}$))

14     $\theta_{final} \leftarrow (\theta_{model}, \theta_{gradient})$ // concatenate $\theta_{model}$ and $\theta_{gradient}$

15     $F(\theta), d(\theta) \leftarrow$ evaluate($\theta_{final}$) // evaluate to get reward $F$ and descriptor $d$

16     $\Theta_{j+1} \leftarrow$ update_population($\theta, F(\theta), d(\theta)$)

17     $q_\phi \leftarrow$ update_dynamics_model($q_\phi$), $Q_\psi \leftarrow$ update_critic($Q_\psi$)

18 **return** $\Theta_J$

---

The transition probabilities $\mathcal{P}(s_{t+1}|s_t, a_t)$ of the environment are usually assumed to be unknown. Model-based RL (Wang et al., 2019b) methods learn a parametric model $p_\phi(s_{t+1}|s_t, a_t)$ typically using supervised learning, from data collected when interacting in the environment. Policies are then trained using transitions obtained by rolling out the model.

## 2.2 QUALITY DIVERSITY

Quality-Diversity (QD) (Pugh et al., 2016; Cully & Demiris, 2017) are diversity-seeking population-based approaches to learning. QD methods maintain a diversity of policies in the population $\Theta$ while maximizing the performance of each policy $\theta \in \Theta$. The population usually contains thousands of policies. QD considers an objective function $F(\theta)$ acting on the parameters of the policy $\theta$. Additionally, QD also considers a behavior descriptor $d(\theta)$ that characterizes the behavior induced by a policy. $d(\theta)$ is used to maintain solutions in their behavioral niche to guarantee the population diversity and that there is no two solutions in the population with similar behavior descriptor $d(\theta)$. When applying QD to a RL problem, they are defined as follows, where $d(\tau)$ is the behavior descriptor of a given trajectory $\tau$:

$$F(\theta) = J(\pi_\theta) = \mathbb{E}_{\tau \sim \pi_\theta}[\mathcal{R}(\tau)] \quad \text{and} \quad d(\theta) = \mathbb{E}_{\tau \sim \pi_\theta}[d(\tau)] \tag{2}$$

Similar to Evolutionary Strategies (ES) (Salimans et al., 2017), QD methods operate on entire episodes and hence both the objective and the descriptor can be computed simultaneously for any parameter vector $\theta$. QD methods then aim to maximize:

$$\max_{\Theta} \quad \text{QD-Score}(\Theta) = \sum_{\theta \in \Theta} F(\theta) \tag{3}$$

By maintaining a population $\Theta$ of both diverse and high-performing policies, QD uses the existing parameters in the population as stepping stones (Nguyen et al., 2016; Wang et al., 2019a) in the optimization. At each iteration $t$, a random set of policies are sampled from the current population $\Theta_j$, perturbed and evaluated in the environment. Based on the results of these evaluations, these perturbed policies might replace existing ones or fill in a new niche in the population. This incrementally improves the QD-Score($\Theta_j$) and encourages creativity and exploration.

## 3 METHOD

In this section, we introduce our two new model-based QD methods: DA-QD-ext and GDA-QD.

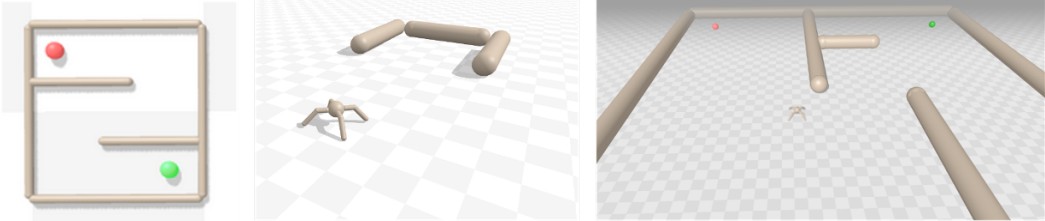

Figure 2: Deceptive reward tasks in the form of PointMaze, AntTrap and AntMaze environments.

### 3.1 DA-QD-EXT: LEARNING AND SIEVING IN IMAGINATION

QD approaches are conventionally driven by random perturbations of the parameters of solutions. This undirected and divergent process gives QD its exploration capabilities but its major drawback is the number of samples it requires. This is especially evident for high-dimensional optimization problems such as optimizing deep neural networks where usually more directed gradient based methods are used. In this work, inspired by Lim et al. (2021), we perform the perturbation process in imagination, relying on a learned dynamics model of the environment to reduce the number of environment interactions when evaluating such perturbed policies.

At each iteration $j$ of the algorithm, the current population $\Theta_j$ is first "copied" into imagination $\tilde{\Theta}_j$. The policies of this provisional population $\pi_{\tilde{\theta}} \in \tilde{\Theta}_j$ are then perturbed, and evaluated in imagination using the rollouts of the dynamics model $q_\phi$. Both the objective $F(\pi_{\tilde{\theta}})$ and the descriptor $d(\pi_{\tilde{\theta}})$ of the policies can be obtained from the state information present in the rollouts. Using this process, $\tilde{\Theta}_j$ undergoes multiple steps of QD optimization in imagination. The resulting policies $\pi_{\tilde{\theta}} \in \tilde{\Theta}_j$ that are added to the provisional population $\tilde{\Theta}_j$ during learning in imagination are then evaluated in the environment and used to update the population $\Theta_j$ if they improve the QD-Score of the population. The updated population $\Theta_j$ is then used as a start for the next iteration of the algorithm. This process of performing QD in imagination acts as a sieve and filters out perturbed solutions that are not likely to improve the quality and diversity of the population, hence increasing the sample efficiency.

Following Chua et al. (2018), we use a probabilistic bootstrap ensemble of models $q_\phi$ to capture uncertainties. Each model in the ensemble is a probabilistic model which predicts parameters of a Gaussian distribution $N(\mu_\phi(s_t, a_t), \Sigma_\phi(s_t, a_t))$ which we can then sample from, capturing the aleatoric uncertainty. This model-based QD method corresponds to DA-QD (Lim et al., 2021). However, we optimize high-dimensional closed loop neural network policies in complex exploration domains and hence, refer to it as DA-QD-ext.

### 3.2 GDA-QD: INCORPORATING GRADIENTS IN QUALITY-DIVERSITY

As mentioned above, the random perturbations driving QD approaches prove inefficient when applied to high-dimensional search spaces such as the parameters of deep neural networks (Colas et al., 2020). To deal with this, we augment the usual perturbation operator with policy gradient information as done by Nilsson & Cully (2021). The policy gradient can be more intuitively thought of as a more directed perturbation, hence being a more efficient optimization update procedure. To apply policy gradients to a populationx, we maintain a critic network $Q_\psi$ which approximates the action-value function $Q(s_t, a_t) = \mathbb{E}\left[\sum_{k=0}^{T-t} \gamma^k r_{t+k+1} \mid s_t, a_t\right]$ and gives the expected return from being in state $s_t$ and following action $a_t$. The critic allows us to gradually improve any policy in the direction maximizing the expected return by computing policy-gradient in Equation 4, approximated over a batch of transitions. We train $Q_\psi$ using the same procedure as TD3 (Fujimoto et al., 2018).

$$\nabla_{\theta_i} J(\theta_i) = \mathbb{E}_{\mathbf{s}, \mathbf{a} \sim \pi_{\theta_i}} \left[\nabla_{\theta_i} \pi_{\theta_i}(\mathbf{s}) \, \nabla_{\mathbf{a}} Q_\psi(\mathbf{s}, \mathbf{a})\right] \tag{4}$$

In DA-QD-ext explained in Section 3.1, the population of policies $\Theta_j$ is copied in imagination $\tilde{\Theta}_j$ to undergo multiple steps of QD optimization and returns $B$ policies to be evaluated. In GDA-QD, we combine the efficient parameter based perturbation in imagination from DA-QD-ext with more directed policy gradient updates explained above. The former is critical for efficient exploration,

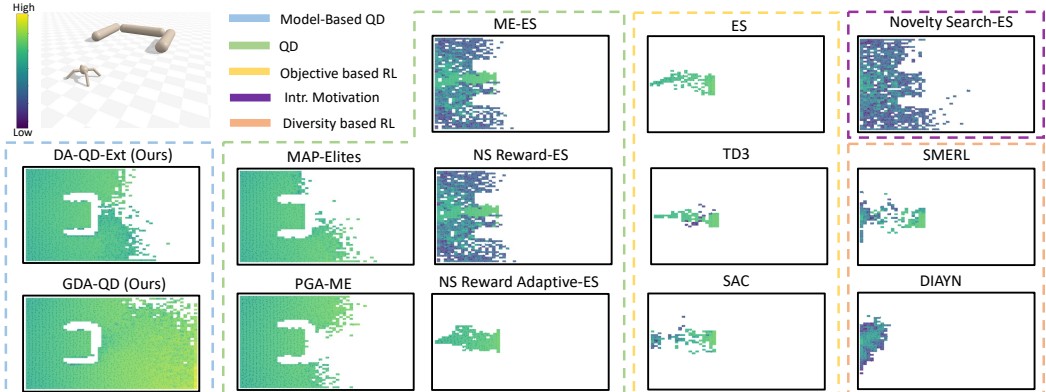

Figure 3: Final population of methods in the AntTrap environment (top left). Each policy in the population is represented with a dot in the final position it manages to reach within the episode. The color of the square around each policy indicates its total reward, the lighter the better. The obstacle clearly appears on this plot, as the empty area in the middle. Our approach GDA-QD does not get stuck in the trap and clearly outperforms baselines given the same number of evaluations.

while the latter is critical for efficient exploitation. To achieve this, a proportion $p_{model}$ of $B$ is sampled to be evaluated directly while another $p_{gradient} = 1 - p_{model}$ is sampled to be improved using the policy gradient operators before being evaluated. In the following, we use $p_{model} = 0.9$. We study the impact of this value and illustrate the complementary properties of these two generation-procedures in Appendix B. We demonstrate that both parameter update procedures are essential to guarantee performance. Figure 1 and Algorithm 1 provides a summary of this algorithm.

A key design choice in GDA-QD for simplicity is that only parameter-based perturbations are applied in imagination, while the policy gradient updates are not applied in imagination. This was to clearly make a separation between efficient exploration and efficient exploitation (through gradient optimization). Applying policy gradient updates in imagination could potentially further improve GDA-QD at the cost of training the critic more often in imagination. We leave this for future work.

### 3.3 QUALITY-DIVERSITY AS DATA GENERATORS FOR DEEP MODELS

A key quality of QD algorithms is that they are excellent at generating diverse and high-quality data (Ecoffet et al., 2021; Grillotti & Cully, 2022). In our work, the search for a diversity of high-performing behaviors when optimizing the population results in a diverse dataset of transitions. We leverage this property to train deep models which require and often excel when provided with such data. As commonly done, the diverse dataset of transitions is stored in a replay buffer and used to train (1) the dynamics model $q_\phi$ and (2) the critic $Q_\psi$. Both these models are suitable candidates as the training of the critic $Q_\psi$ is off-policy and can trained using transitions collected from any behavioral policy. Additionally, training the dynamics model $q_\phi$ is a supervised learning problem which would benefit from a large and diverse dataset of transitions.

We found this property to be especially important to our method. Specific to the training of the dynamics model, GDA-QD does not use transitions produce by gradient-optimized policies to train the model. This was found empirically in our studies as we observed the transitions produced by the gradient-optimized policies induced a shift in distribution that prove detrimental for the training of the dynamics models. Results of this study are detailed later in Section 4.3. It is important to recognize that the transitions used to train the dynamics model are not merely just from policies that have been randomly perturbed but a population of policies that have undergone multiple steps of QD optimization in imagination and are expected to improve the population. This results in a diverse and high performing dataset of transitions that GDA-QD uses to further train its models.

## 4 EXPERIMENTS

We aim to evaluate our method by answering four main questions: (1) Can we scale model-based QD approaches to RL domains and Neuroevolution? (2) Does GDA-QD results in more performant

| | PointMaze | | | AntTrap | | | AntMaze | | |
| --- | --- | --- | --- | --- | --- | --- | --- | --- | --- |
| | QD-Score | Cov | Max-Rew | QD-Score | Cov | Max-Rew | QD-Score | Cov | Max-Rew |
| TD3 | - | - | -126.38 | - | - | 189.52 | - | - | 1.05 |
| SAC | - | - | -126.18 | - | - | 204.68 | - | - | 1.06 |
| ES | 0.46 | 0.52 | -126.85 | 2.97 | 2.91 | 200.95 | 18.6 | 10.88 | 0.97 |
| DIAYN | - | - | -67.98 | - | - | -6.15 | - | - | 0.20 |
| SMERL | - | - | -38.29 | - | - | 171.81 | - | - | 1.06 |
| NS-ES | 0.93 | 1.8 | -147.80 | 10.12 | 28.76 | -13.14 | 45.45 | 42.6 | 1.23 |
| ME | 93.74 | **99.92** | -25.36 | 42.44 | 43.44 | 218.77 | 56.71 | 37.68 | 1.29 |
| PGA-ME | 93.06 | **99.92** | **-24.06** | 47.82 | 47.08 | 274.52 | 62.56 | 39.66 | 1.48 |
| NSR-ES | 1.07 | 1.24 | -126.85 | 6.36 | 6.26 | 196.30 | 22.89 | 14.24 | 1.02 |
| NSRA-ES | 1.44 | 1.78 | -126.85 | 14.18 | 29.32 | 170.38 | 46.5 | 43.56 | 1.30 |
| ME-ES | 21.0 | 30.52 | -62.05 | 12.70 | 21.92 | 157.42 | 38.5 | 33.48 | 1.15 |
| DA-QD-ext | 96.67 | **99.92** | -24.81 | 50.33 | 51.0 | 196.66 | 70.94 | 43.7 | 1.51 |
| GDA-QD | **97.70** | **99.92** | -24.24 | **76.28** | **72.44** | **342.24** | **80.5** | **51.4** | **1.87** |

Table 1: Final QD-Score (% of maximum value), Coverage (%) and Max-Total-Reward reached by all algorithms on all considered tasks. Each experiment is replicated 15 times, we report in the table the median value across runs. In the algorithms name, ME stands for MAP-Elites.

and sample-efficient learning than traditional QD approaches and simple model-based QD? (3) What is the importance of the policy-gradient perturbation in the performance of GDA-QD? (4) How does the data-generation capabilities of GDA-QD enforce efficient learning?

## 4.1 Experimental setup

**Tasks and Environments:** We focus on tasks considered in literature as hard exploration problems: PointMaze Lehman & Stanley (2011b); Parker-Holder et al. (2020), AntTrap (Conti et al., 2018; Colas et al., 2020; Cideron et al., 2020; Parker-Holder et al., 2020) and AntMaze (Colas et al., 2020; Cideron et al., 2020; Salter et al., 2022) (see Fig. 2). The reward in these tasks is deceptive making exploration and diversity critical when solving them. To start, we consider a simple PointMaze environment where a 2-dimensional point agent is given a reward corresponding to the distance to the goal in the maze. The AntTrap and AntMaze are higher dimensional continuous control tasks where an 8-DoF Ant robot learns how to walk, aiming to go beyond the trap in AntTrap and to reach the goal in AntMaze. In AntTrap, the robot gets increasing rewards for going as fast as possible while minimizing energy-usage. In the AntMaze tasks, the reward is the distance to the goal. This reward definition for all the tasks considered makes them deceptive. The descriptor $d(\pi_\theta)$ used in all the tasks is defined as the x-y position at the end of the trajectory $(x_T, y_T)$.

**Baselines:** Across our experiments, we consider the following baselines:
- **MAP-Elites:** the most-commonly used QD algorithm (Mouret & Clune, 2015).
- **PGA-MAP-Elites:** (Nilsson & Cully, 2021), augments MAP-Elites with a policy-gradient based update operator.
- **OpenAI ES:** Evolution Strategy (Salimans et al., 2017) relying on natural gradient approximation.
- **Novelty Search ES:** We compare against NS-ES Conti et al. (2018) as an intrinsic motivation baseline. This uses the OpenAI-ES algorithm but with a novelty reward instead of task reward..
- **QD-ES Algorithms:** QD-ES algorithms consider both quality and novelty during optimization. We use NSR-ES, NSRA-ES (Conti et al., 2018), and MAP-Elites-ES (ME-ES) (Colas et al., 2020). NSR-ES and NSRA-ES build on NS-ES by including the task reward term as a weighted sum with the novelty reward term. ME-ES mixes MAP-Elites with OpenAI ES (Colas et al., 2020).
- **Single Policy Deep RL Algorithms:** We consider TD3 Fujimoto et al. (2018) and SAC Haarnoja et al. (2018). SAC is entropy regularized and is a popular choice for greater exploration.
- **Mutual Information RL Algorithms:** We also consider DIAYN Eysenbach et al. (2018) and SMERL Kumar et al. (2020) which are also diversity seeking algorithms. To ensure the comparisons are fair with descriptor based methods, we use the x-y prior when running these algorithms. DIAYN is purely unsupervised and does not consider the task rewards. SMERL considers the task reward during optimization

For fairness, as the single policy baselines (TD3, SAC, DIAYN, SMERL) do not rely on a population, nor on complete-episode evaluations, we only reports its final value in the results as a dotted

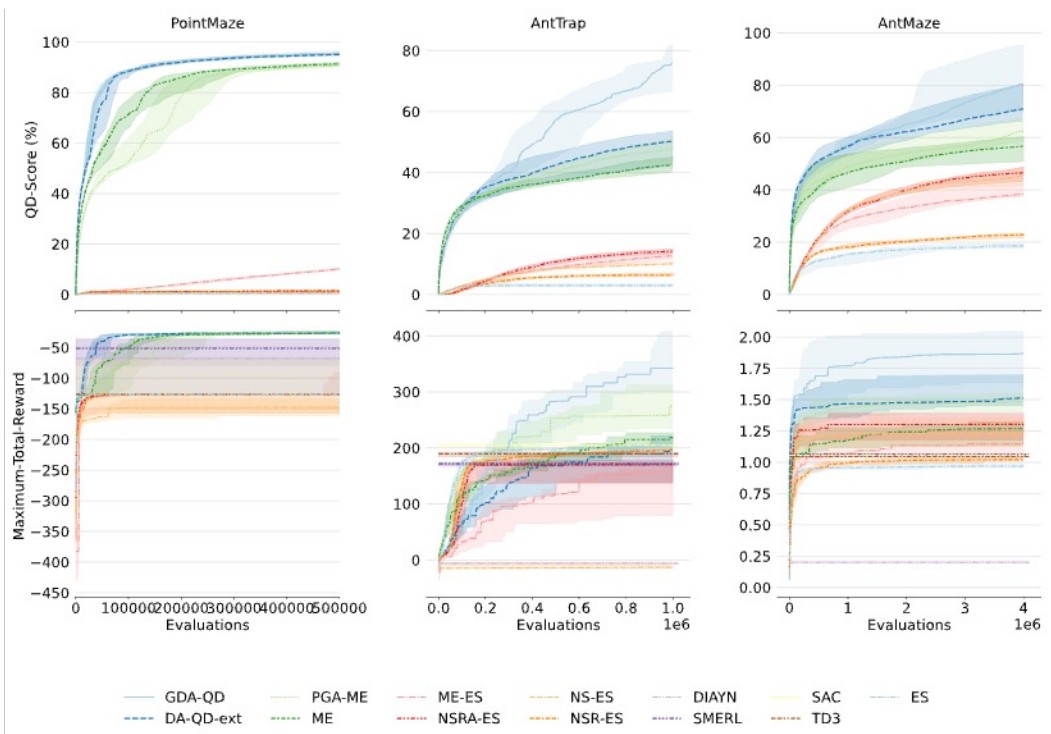

Figure 4: QD-Score (top) Max-Total-Reward (bottom) of all algorithms on the AntTrap (left), PointMaze (centre) and AntMaze (right) tasks plotted against number of evaluations. Each experiment is replicated 15 times, the solid line corresponds to the median over replications and the shaded area to the first and third quartiles.

line. For a more qualitative comparison of these algorithms, we also collect the trajectories of the policies throughout the learning process and plot them as part of a population. To ensure the comparability of algorithms using ES (OpenAI-ES, NS-ES, NSR-ES, NSRA-ES, ME-ES) with other approaches, we consider every estimate-evaluation as one sample, making these algorithms highly sample-inefficient.

**Metrics:** We consider two metrics to assess the performance of GDA-QD:
- **QD-Score:** defined in Section 2.2. It quantifies the diversity and quality of the overall population and allows to compare population-based methods.
- **Max-Total-Reward:** the total reward of the best individual of the current population. This metrics allows comparison with single-policy methods such as RL baselines.

**Implementation and hyperparameters:** Our source code is available at TO-BE-RELEASED. [1]. It includes a containerised environment to replicate our experiments. The methods presented in this paper as well as all our baselines are based on the implementation of MAP-Elites in the QDax open-source library (Lim et al., 2022), using the Brax simulator Freeman et al. (2021). All hyperparameters and implementation details used in our algorithms and for model training can be found in the Appendix A.

## 4.2 RESULTS

The results of our experiments are summarized in Figure 4 and Table 1. We also display a visualization of the final populations of policies for each algorithm in Figure 3 (AntTrap) and Appendix C (PointMaze and AntMaze).

Figure 4 shows that both our proposed model-based versions, DA-QD-ext and GDA-QD, significantly outperform all baselines in terms of sample efficiency and final performance. This demon-

---

[1]The code will be made available after the review period

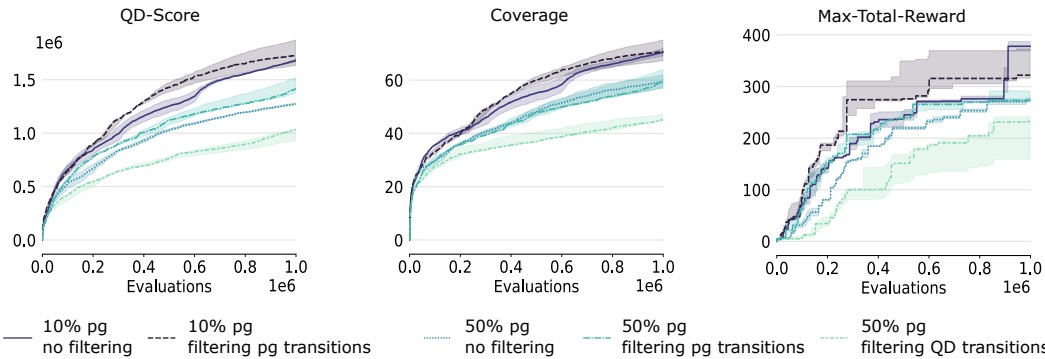

Figure 5: QD-Score (left), Coverage (middle) and Max-Total-Reward (right) on the AntTrap task of GDA-QD with different proportion of policy-gradients generated policies (pg) and different transitions filtering schemes. The solid line corresponds to the median over replications and the shaded area to the first and third quartiles.

strates that we can scale model-based QD methods to deep neuroevolution. The performance of DA-QD-ext suggests that learning diverse behaviors in imagination using the dynamics model is a simple but effective approach to save samples yet maintain the divergent search capabilities of the random perturbations. However, we can see that the maximum total reward obtained by DA-QD-ext seems to stagnate and increase slowly, especially when compared to GDA-QD and PGA-MAP-Elites. This can be explained by the absence of reward maximizing optimization updates such as the policy gradient updates present in GDA-QD and PGA-MAP-Elites. Hence, GDA-QD is shown to get the "best of both worlds" by benefitting from the QD in imagination as well as the policy gradient updates. This is evident in its performance across the QD-Score and max. return metrics.

In terms of baselines, Figure 3 show that all objective-only baselines: SAC, TD3 and ES, struggle with deceptive rewards. They all get stuck into the traps present in AntTrap and AntMaze. Diversity Based RL algorihtms such as DIAYN struggle with no task reward signal while SMERL manages to reach the final goal only in the simpler PointMaze. The single-agent baselines (TD3, SAC, DIAYN, and SMERL) are plotted as horizontal lines to represent the max performance obtained by the agent. This is done to allow comparison as they do not rely on a population, making the Coverage and QD-Score metrics not relevant, and also performs policies-updates within episodes. The poor performance of ES-based algorithms, in particular ME-ES, can be attributed to the number of samples ($\sim$ hundreds or thousands) required just to approximate a single gradient step. It is important to note that these methods commonly do not consider the notion of evaluations and generally evaluate the algorithms versus time or number of generations as they are suited to be heavily parallelized across clusters of CPUs. Despite this limitation, intrinsically-motivated ES baselines NS-ES and its QD variant NSRA-ES manage to get good coverage on AntTrap and AntMaze. However, they struggle to discover high-performing solutions within the given evaluation budget. We provide a visualization of the adaptive mechanism of NSRA-ES in Appendix C.

### 4.3 IMPORTANCE OF QD FOR TRAINING DEEP MODELS

To investigate the data generation abilities of QD, we compare the performance of the algorithm when the dynamics model is trained on different data distributions based on the policies rolled out in the environment. In our case, we have two main types of data generators: policies obtained thought the QD process in imagination (giving $D_{model}$), and policies perturbed also using policy-gradients (giving $D_{gradient}$). We run an ablation where we train the dynamics model on transitions collected by either $D_{model}$, $D_{gradient}$, or a mixture of both $D_{model+gradient}$, by filtering out the corresponding transitions. Figure 5 shows the performance curves when running this ablation on the AntTrap task. We first test the effect of training the model with $D_{model}$ when $p_{gradient} = 0.1$. We notice a minor difference in which experiments that train only on $D_{model}$ perform better than when not using any filter (i.e. $D_{model+gradient}$). To enable a fair comparison in terms of the number of transitions added to the replay buffer when attempting to do the converse (i.e. using $D_{gradient}$), we use a $p_{model} = p_{gradient} = 0.5$. We observe better performance when the dynamics model is only trained with $D_{model}$. This is compared to training on $D_{model+gradient}$ where the difference

is minimal as seen previously. However, performance significantly drops when training just on $D_{gradient}$. We hypothesize that this is due to bias in the transitions obtained through the policies perturbed by policy gradient resulting in a skewed dataset while the transitions given by policies obtained from QD in imagination provides a diverse and high-quality dataset.

## 5 RELATED WORK

**Searching for Diversity.** Prior studies have shown the importance of maintaining a diverse set of solutions to solve a problem. Novelty-search approaches (Lehman & Stanley, 2011a) inspired from evolutionary computation optimize for the novelty of solutions defined by a behavioral characterization with respect to a population instead of the optimization objective. QD methods (Cully & Demiris, 2017; Mouret & Clune, 2015) extend this approach by also considering the objective, aiming to find both diverse and high-performing population of solutions. Similarly, our work builds on QD approaches with the aim to maintain a diversity of high-performing solutions. Other approaches to searching for diversity also exist in the RL community. Unsupervised RL methods (Eysenbach et al., 2018; Sharma et al., 2020) commonly use a mutual information maximization objective to learn a diversity of behaviors in a skill-conditioned policies. Similar to QD, Kumar et al. (2020); Zahavy et al. (2022) have proposed to extend these unsupervised RL approaches by integrating objectives using constrained Markov decision processes. While these approaches typically focus on maintaining a dozen of different policies, our algorithm discovers thousands of independent diverse policies.

**Neuroevolution** seeks to evolve neural networks through biologically-inspired methods such as evolutionary algorithms and have interesting properties unavailable to common gradient-based methods (Stanley et al., 2019). However, a limitation in the neuroevolution domain is the dimensionality of the search-space, that quickly limits the effectiveness of random perturbations. Some methods have overcome this limitation by using indirect encoding methods (Clune et al., 2009) or through natural approximated gradients (Salimans et al., 2017). Our work hybridizes neuroevolution with deep reinforcement learning methods. Similar to Nilsson & Cully (2021), we use policy-gradients as directed perturbations to effectively maneuver the high-dimensional search space but, we significantly improve the sample efficiency and performance by also augmenting these operators with model-based methods.

**Model-based Quality-Diversity.** As the perturbations commonly used in QD are sample inefficient, prior work has sought the use of data-driven models to alleviate this. SAIL Gaier et al. (2018) introduced the use of a surrogate model in the form of a Gaussian Process model to predict the objective. As Gaussian processes generally only work well on low-dimensional data, more recent methods have explored the use of deep networks (Keller et al., 2020; Lim et al., 2021; Zhang et al., 2022) as forms surrogate models to predict both the objective and descriptors. We utilize the model-based QD framework from Lim et al. (2021) which first introduced the idea of maintaining an imagined population and also builds on model-based RL methods (Wang et al., 2019b) where a dynamics model is used. Critically, this work has only been applied to low-dimensional open-loop policies. To the best of our knowledge, our work is the first model-based QD algorithm that scales to the more complex deep neuroevolution domain where we optimize closed loop RL policies.

## 6 CONCLUSION AND FUTURE WORK

In this paper, we introduce a novel model-based Quality-Diversity method, GDA-QD, which optimizes a population of diverse policies to explore more efficiently. To the best of our knowledge, this approach is the first model-based QD algorithm scaling to neuroevolution to optimize deep neural network policies. We leverage a key property of QD algorithms as effective data generators to train deep models in the form of a dynamics model and a critic. In turn, these models help to significantly improve the sample efficiency and final performance of the QD algorithm. The dynamics model is used to learn and sieve policies in imagination while the critic is used to apply policy gradient updates to sampled policies. Our experiments show that GDA-QD outperforms a range of Deep RL and QD baselines on a hard exploration task containing deceptive rewards. Overall, we demonstrate some of the powerful synergies that can arise between population-based learning and deep learning approaches. In future work, we hope to extend our work to more complex domains through the use of latent dynamics models (Ha & Schmidhuber, 2018; Hafner et al., 2019).

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
