# OpenReview forum: "Efficient Exploration using Model-Based Quality-Diversity with Gradients"
_ICLR.cc/2023/Conference — Submitted to ICLR 2023_

### Official Review · Reviewer_QkBZ · 2022-10-24

**Confidence:** 3
**Clarity, Quality, Novelty And Reproducibility:** sound
**Correctness:** 3
**Technical Novelty And Significance:** 3
**Empirical Novelty And Significance:** 2
**Recommendation:** 5

**Strength And Weaknesses:**

Strength:

The idea is simple and straight, the experiment results seem significantly improves the sample efficiency and quality of solutions to the RL QD problem.

Weaknesses:

(1) It is not clear how the dynamics model is built. Although the author has mentioned that they apply a probabilistic bootstrap ensemble of models, the reviewer is still confused about the method, goal, and other details for training the dynamics model.

(2) The experimental results are limited to a simple AntTrap environment. More comparisons of other environments are expected.

(3) The training curve in Figure 2 and Figure 4 should take the overhead of training dynamics models into account since they perform some level of feature extraction and generalization. A fair comparison would be setting the total training time as the x-axis when using the same hardware resources.


**Summary Of The Paper:**

This paper proposes a Quality-Diversity approach that bootstraps the sample efficiency with a learned dynamics model. The authors demonstrate that the model-based QD method outperforms both Deep RL and QD baselines in a hard-exploration task.
The idea is simple and straight, but the writing of the paper should be improved since the reviewer hardly follows the motivation and the components in Figure1. Besides, the experimental results are quite limited to a simple environment. The paper pays much content to the ablation study, which is trivial.

**Summary Of The Review:**

see above

---

### Official Review · Reviewer_TxSP · 2022-10-24

**Confidence:** 4
**Correctness:** 3
**Technical Novelty And Significance:** 2
**Empirical Novelty And Significance:** 2
**Recommendation:** 3

**Clarity, Quality, Novelty And Reproducibility:**

### Clarity
Several important details are missing:
- As mentioned previously, the paper can benefit from making it clearer which aspects of this work are original contributions with respect to prior works.
- The hyperparameter table in the Appendix is hard to read (symbolic notation is not used) and there is no description of the numerous hyperparameters listed.
- The key design choice of only performing gaussian perturbations (as opposed to using PG updates) in the imagined updates in the dynamics model is not explained.
- The values in the QD-score column in Table 1 seems hard to interpret. It would make more sense to compare the row values here in terms of a normalized or relative QD score, since the exact values and units here in themselves are not particularly meaningful.
- Figure 1 is hard to understand, as the number labels follow an erratic pattern through the diagram and several variables like $f$ and $d$ are not explained in the diagram. Further it is not clear what the icon next to $d$ represents.
- The abstract should make it clearer that the method extends a QD method with policy gradient updates. Currently the algorithm description in the abstract is vague.
- The authors could consider using different colors to communicate both DA-QD and GDA-QD in Algorithm 1, which would make it much easier to see exactly how the two versions of the algorithm differ.

### Quality
- The overall writing and figure descriptions can benefit from a copy-editing pass.

### Reproducibility
- The hyperparameter table in the appendix is unclear and no code is provided. Thus reproduction of the experiments in this paper would likely be challenging.

### Novelty
- This paper seems to present an extension of existing works using dynamics-aware QD for RL. Given that the algorithmic contribution is incremental, this work would benefit from a more comprehensive series of experiments comparing to alternative methods and clearly controlling for the various components of the method.

**Strength And Weaknesses:**

### Strengths
- The background concepts are clearly motivated and communicated.
- Combination of TD3 with DA-QD seems to be a novel algorithmic configuration.
- The presentation of solution populations for each method in Figure 3 is well done.

### Weaknesses

**Experimental setting seems inappropriate for benchmarking QD**

Most importantly, the method is based on QD, but the experimental setting is an exploration problem that does not necessarily require QD. I recommend using a different experimental setting that focuses on the benefits of QD over exploration. In particular, QD inherently maintains a population of solutions, making such methods advantageous for multi-modal optimization. Thus, an experimental setting that requires learning solutions for a multi-modal solution space, or that requires adapting to changes in the environment, e.g. if the ant's legs can become damaged, would be much more appropriate than Ant-Trap, which is simply an exploration problem solvable with existing exploration methods in RL.

Regarding the Ant-Trap environment in particular: Deep RL has dealt with such problems using exploration methods, such as entropy regularization and intrinsic motivation. Moreover, there exist efficient quality-diversity methods in deep RL, such as DIAYN, DOMiNO, and DvD, which do not require maintaining two populations or the use of a dynamics model. Thus, at a minimum, it seems important for this work to compare to the following:
- SAC [1], which learns a maximum entropy policy.
- DIAYN [2].
- A basic intrinsic motivation method (e.g. RND [3]).

DOMiNO [4] and DvD [5] both require pre-specifying a small population compared to the MAP-Elites methods used in this work, so it may be possible to find settings where their use of relatively smaller populations proves a disadvantage or makes them inapplicable.

There also seems to be another important baseline missing: The current baselines control for the use of dynamics model (PGA-MAP-Elites), the use of dynamics model + TD3 (MAP-Elites), as well as the use of dynamics model + MAP-Elites (TD3), in addition to comparing to ES baselines, but these ES baselines do not seem directly relevant to the study. Importantly, the study seems to be missing a comparison to TD3 with a dynamics model (control for the use of MAP elites).

Lastly, the comparison to vanilla TD3 seems unfair, as the DA-QD methods maintain large populations, while TD3 is effectively a population of size 1. A fair comparison would be to run a population of TD3 agents (e.g. set to the size of the batch size used for MAP-Elites updates), and compare QD-score, coverage, and max total reward with respect to this population.

**Additional weaknesses**
- The paper's presentation of the DA-QD algorithm seems to obfuscate the originality of contributions. It is not clear from the writing whether the DA-QD algorithm itself is a new contribution or simply taken from Lim et al, 2021. This is because the introduction states the paper introduces an extension of DA-QD called DA-QD-ext, but then refers to the algorithm as simply DA-QD in the rest of the paper. Which contributions are new should be clearly stated in the paper.
- DA-QD and GDA-QD seem to require a lot more evaluations than TD3 due to maintaining two large populations inside of the real and imagined MAP-Elites archives. It would be useful to see a comparison of number of steps needed.
- The paper does not motivate the key design choice of only performing gaussian perturbations in the model. It is not clear why PG-based updates are not used when updating the imagined population.

### References
- [1] Haarnoja, Tuomas, et al. "Soft actor-critic algorithms and applications." arXiv preprint arXiv:1812.05905 (2018).
- [2] Burda, Yuri, et al. "Exploration by random network distillation." arXiv preprint arXiv:1810.12894 (2018).
- [3] Eysenbach, Benjamin, et al. "Diversity is all you need: Learning skills without a reward function." arXiv preprint arXiv:1802.06070 (2018).
- [4] Zahavy, Tom, et al. "Discovering Policies with DOMiNO: Diversity Optimization Maintaining Near Optimality." arXiv preprint arXiv:2205.13521 (2022).
- [5] Parker-Holder, Jack, et al. "Effective diversity in population based reinforcement learning." Advances in Neural Information Processing Systems 33 (2020): 18050-18062.

**Summary Of The Paper:**

This paper presents two new algorithms, DA-QD and GDA-QD, for training a large population of diverse reinforcement learning agents for continuous control. These algorithms combine the benefits of surrogate-assisted quality-diversity (QD) and off-policy RL. Specifically, DA-QD maintains two MAP-Elites buffers, one real and one "imagined," both along two trajectory-based behavioral descriptors. At each iteration of the algorithm, the real MAP archive is copied into the imagined MAP archive, and MAP-Elites with gaussian perturbations is applied to the imagined population based on rollouts in the learned dynamics model. The final, new elites after this step are then the candidates to be evaluated on the real environment. These real rollouts are then used to update the real MAP archive as well as update the learned dynamics model. GDA-QD is the same algorithm with the exception that only some proportion $p$ of real rollouts is directly evaluated for inclusion in the real archive, while a proportion $1-p$ is updated with TD3. They show on a continuous control task with a local minimum, Ant-Trap, that GDA-QD's use of a learned dynamics model leads to much higher returns than variants of MAP-Elites with ES and policy-gradient updates, as well as TD3.

**Summary Of The Review:**

This paper presents an incremental extension of a model-assisted QD method by extending it with TD3 policy gradient updates. However, given that incremental nature of the algorithmic contribution, the work should feature a more comprehensive experimental setting. Moreover, the current environment used in the experiments is simply an exploration problem and thus does not provide a useful benchmark for QD. Based on these issues, I cannot recommend this paper for acceptance.

---

### Official Review · Reviewer_FZRr · 2022-10-24

**Confidence:** 4
**Correctness:** 3
**Technical Novelty And Significance:** 2
**Empirical Novelty And Significance:** 2
**Recommendation:** 6

**Clarity, Quality, Novelty And Reproducibility:**

+ The paper is clearly written.
+ The algorithm is novel.

**Strength And Weaknesses:**

+ Clear description of the justification and the proposed technique.
+ Interesting hybridization of reinforcement learning and neuroevolution.
+ Performance increase over other QD approaches over the chosen experimental problem.
- The system positions itself within the QD community, with most of the baseline algorithms coming from the same community.
- The system is only evaluated on the ant trap problem, which is hand engineered to trigger a particular kind of behavior, of the ant being trapped in the obstacle. It is unclear whether the proposed approach would generalize to any other type of environments.

**Summary Of The Paper:**

The paper considers the case of population-based approaches for exploration in reinforcement learning, in particular the Quality-Diversity approach, an evolutionary algorithm to create collections of high performing solutions. The authors propose two algorithm variations of QD which take into consideration the learned dynamics model and the policy gradient. The resulting policy is evaluated on the ant-trap problem, a hard exploration problem with a deceptive reward, where the short term optimization of the reward leads the agent into a trap. The resulting algorithm exceeds the performance and is more sample efficient that the baseline QD approach.

**Summary Of The Review:**

The paper proposes an approach that integrates the quality-diversity approach with reinforcement learning. The proposed approach outperforms other QD approaches on the ant-trap problem, but it is unclear how it generalizes further.
---
The authors had made several additions to the paper during the discussion session, which convinced me to raise my score.

---

### Official Review · Reviewer_tyZo · 2022-10-25

**Confidence:** 3
**Correctness:** 3
**Technical Novelty And Significance:** 2
**Empirical Novelty And Significance:** 2
**Recommendation:** 6

**Clarity, Quality, Novelty And Reproducibility:**

I personally didn't find the paper that clear, but upon several passes, I believe I understand the main ideas presented. I do think the quality is overall fairly good, but I've listed some caveats above. I don't think the work is that novel, as model-based QD algorithms have been explored before for the same purpose as here, and the critic-guided policy updates have also been covered in prior work, although the application of QD + models to deep RL policies seems novel. At the moment the work seems to have low reproducibility (some implementation details are missing, as listed above), but I am encouraged to see the authors are keen to opensource their efforts.

**Strength And Weaknesses:**

Strengths:
* The core idea of augmenting QD with a model makes a lot of sense, whereby costly real-world samples can be avoided by sampling imaginary trajectories instead.
* Results appear promising on the evaluated environment.

Weaknesses:
* I found the paper rather difficult to read, and generally not that well written. I caveat that this may be a consequence of my lack of experience with QD literature, but I found myself requiring several read throughs of the paper before I felt I fully understood what was going on. To give some examples of things I found difficult to parse:
    * In the algorithm box, many variables and functions aren't explained (e.g., $p_{model}$, $q_\phi$, last_added). While they are introduced somewhere in the main body, I found myself having to flit between the body and box quite frequently.
    * Similarly, I found section 3.2 hard to parse and rather terse, specifically the final paragraph. There is quite a bit of terminology introduced and they all seem to interact with each other here. Some repetition, or a clearer approach would be appreciated.
    * The behavior descriptor is mentioned in a few places, but never defined mathematically; it would have been useful to define this either in the appendix or when introduced. It was also not clear to me how the descriptor was combined with the objective function.
    * "This means that we investigate the performance of the algorithm when the dynamics model is trained on different data distributions based on the policies rolled out in the environment.". It's not entirely clear to me what the authors are trying to say here? It's seems rather imprecise (what are the different distributions? Why these different distributions (i.e., pg v.s. non-pg)?), and made it hard for me to understand the motivation of this section.
* The quality of the experiments isn't good enough; only one environment is presented, and only 3 seeds. While I appreciate population based methods might be expensive to run, given the lack of environments, it seems reasonable to expect more seeds.
* The result of filtering out the PG trajectories for model-training seems counterintuitive to me, why is it the case that the performance is lower when including more data? The only explanation given in the paper is that the data are "skewed", and I think this could merit more investigation as it's counter intuitive. I think something showing MSE/Log-likelihood on a held out test set of dynamics could shed light here, or perhaps t-SNE/traces of imagined trajectories.
* I'd like some clarity on
    * What is greedy learning rate in the appendix?
    * What horizon was used in the model rollouts?
    * What is the sample efficiency of TD3 v.s. QD algorithms in terms of environment steps?
* Nits:
    * The TD3 and OpenAI ES citations are mixed up
    * In Algorithm 1, typo in `evaluate_imgination`
    * withing -> within

**Summary Of The Paper:**

In this work, the authors introduce a model-based quality diversity (QD) approach, which aims to both solve a problem whilst inducing diverse solutions. To do this, the authors perform several rounds of quality diversity optimization over their population inside the world model, and further augment a subset of the population with a policy gradient from the critic post-hoc, with the aim of guiding solutions towards those that solve the underlying MDP (i.e., the quality part of QD).

They show that by augmenting QD approaches with a model, they achieve improved sample efficiency (by performing the usual costly rollouts inside a world model), and also strong asymptotic performance. They also show findings regarding the impact of world model data distribution and policy gradient proportion on final performance.

**Summary Of The Review:**

Overall the idea of combining model-based approaches with QD algorithms is nice, and this work lends further support to this direction of research. However I have issues regarding the presentation, analysis and experiments which means I am not comfortable recommending acceptance at this point. I'd be happy to raise my score if the authors are able to address these.

---

The authors largely addressed my concerns, hence I've increased my score.

---

### Comment · Area_Chair_LMmQ · 2022-11-18
**Responses**

Dear Reviewers,

Do you have any comments/replies to author's responses - it would be great if you could respond to them. Have they changed your opinion on the paper?

Kind regards,
AC

---

### Decision · Program_Chairs · 2023-01-20

**Decision:**

Reject

**Justification For Why Not Higher Score:**

Not a sufficient experimental evaluation.

**Justification For Why Not Lower Score:**

N/A

**Metareview: Summary, Strengths And Weaknesses:**

This paper combines quality diversity approach with improving policies using gradients and imagination in a model to update population. While pieces of this have been studied before, this combination is novel.
Putting these together, does not constitute a fundamentally novel method, and therefore to be justified, it needs to have a strong experimental validation. The main drawback of this paper is testing on a rather toy environments that are not sufficiently convincing that the method is useful in general.